# Detecting epinephrine auto-injector shortages in Finland 2016–2022: Log-data analysis of online information seeking

Milla Mukka[1]*, Samuli Pesälä[1], Pekka Mustonen[2], Minna Kaila[3], Otto Helve[4,5]

1 University of Helsinki, Helsinki, Finland, 2 Duodecim Publishing Company Ltd, Helsinki, Finland, 3 Clinicum, University of Helsinki, Helsinki, Finland, 4 Department of Health Security, Finnish Institute for Health and Welfare, Helsinki, Finland, 5 Paediatrics Research Center, Children's Hospital, Helsinki University Hospital and the University of Helsinki, Helsinki, Finland

* milla.mukka@helsinki.fi

**Data Availability Statement:** All relevant data are within the paper and its Supporting Information files.

## Abstract

### Introduction

Medicine shortages prevail as a worldwide problem causing life-threatening situations for adults and children. Epinephrine auto-injectors are used for serious allergic reactions called anaphylaxis, and alternative auto-injectors are not always available in pharmacies. Health-care professionals in Finland use the dedicated internet source, Physician's Database (PD), when seeking medical information in practice, while Health Library (HL) provides health information for citizens (S1 Data). The objectives were to assess whether (1) professionals' searches for epinephrine auto-injectors and (2) citizens' anaphylaxis article openings relate to epinephrine shortages in Finland.

### Methods

Monthly log data on epinephrine auto-injectors (EpiPen®, Jext®, Emerade®) from PD and on openings of anaphylaxis articles from HL were collected during 2016–2022. Profession-als' searches of seven auto-injectors and citizens' openings of four anaphylaxis articles were compared to information on epinephrine shortages reported by Finnish Medicines Agency. Professionals' auto-injector prescriptions provided by Social Insurance Institution were also assessed.

### Results

Total searches in EpiPen® (N = 111,740), Jext® (N = 25,631), and Emerade® (N = 18,329) could be analyzed during 2016–2022. EpiPen® only could visually show seasonal patterns during summertime, peaking vigorously in the summer of 2018 when the major EpiPen® shortage appeared worldwide. Anaphylaxis articles equaled 2,030,855 openings altogether. Openings of one anaphylaxis article ("Bites and Stings") peaked during summertime, while another article ("Anaphylactic Reaction") peaked only once (three-fold increase) at the end of 2020 when COVID-19 vaccinations started, and auto-injector prescriptions were lowest.

**Funding:** Open access funded by Helsinki University Library

**Abbreviations:** ATC, Anatomical Therapeutic Chemical code; COVID-19, Coronavirus disease 19; Fimea, Finnish Medicines Agency; HCPs, healthcare professionals; HL, Health Library; Kela, Social Insurance Institution; PD, Physician's Database.

Fifty EpiPen®, one Jext®, and twelve Emerade® shortages were reported. Almost a two-fold increase in peaks of auto-injector prescriptions was found during summertime.

## Conclusion

This study shows that (1) epinephrine shortages related to professionals' searching for auto-injectors, and (2) citizens' information seeking on anaphylaxis related to summertime and shortages with lesser prescriptions. Therefore, the dedicated internet databases aimed at professionals and citizens could be used as additional information sources to detect anaphylactic reactions and auto-injector shortages.

## Introduction

### Medicine shortages

Medicine shortages are a growing worldwide issue [1], influencing low-, middle-, and high-income countries worldwide [2]. Even if manufacturing, distribution, and transport technologies have greatly developed [3], the causes of medicine shortages are usually complex, including raw material and manufacturing issues, logistic and regulatory problems, as well as different seasonal demands [2]. However, the reasons for some shortages remain unknown. Along with clinical, economic, and humanistic impacts on patient outcomes [4], medicine shortages may also result in inadequate patient care and high expenses for institutions [5]. A scoping review found that patients reported more increased costs, medication errors, adverse events, mortality, and complaints during shortages [4]. To minimize the problems related to shortages, some approaches have been suggested, such as alerting physicians on shortages and providing further information on alternative products [6].

Finland, located in Northern Europe, is a geographically large but sparsely-populated country where medicine shortages stem from these country-specific characteristics, including the small size of the pharmaceutical market, fluctuating demand, small stock size, long delivery time, and complex production chain [7]. In Finnish community pharmacies, medicine shortages are common, meaning that 79.8% of pharmacies have reported daily or almost daily medicine shortages [8]. The wholesaler has reported the reason for the shortage to community pharmacies in 11.2% of medicine shortage cases. 67.0% of pharmacies reported that medicine shortages did not cause problems for pharmacies, since an alternative (i.e., substitutable) product was available in 48.5% of pharmacies. However, medicine shortages cause customer dissatisfaction and increase the workload of professionals in community pharmacies [8].

### Epinephrine auto-injectors

Epinephrine (adrenaline) is used as first aid for serious allergic reactions called anaphylaxis [9]. There are many causes of anaphylaxis in adults and children, such as venom from biting or stinging insects, food, and medication [9]. Many seriously-allergic patients carry epinephrine auto-injectors (self-administrable and injectable medications) with them since anaphylaxis might be life-threatening if an auto-injector is not rapidly available. Studies have suggested that epinephrine auto-injectors should be available in community spaces, such as schools [10, 11], and people may also carry expired epinephrine auto-injectors with them [12]. There are three epinephrine auto-injectors available in Finland, namely EpiPen®, Jext®, and Emerade®. The lack of availability of epinephrine auto-injectors has raised concerns

worldwide for individuals of all ages [13], and the demand for alternative products to Epi-Pen® was such that they were rapidly sold out in Finnish pharmacies in 2018 [14]. Epinephrine is on the WHO list of essential medicines [15].

## Online information seeking

Professionals in various medical sectors in Finland may search for information on medicine shortages. Infodemiology is defined as the area of online information distribution aimed at informing public health and public policy [16], playing an important role in understanding health-related behavior [16] and health informatics research [17]. Little data exist on how HCPs and citizens seek online medical information during medicine shortages. Google Trends has been used as an early indicator of intravenous immunoglobulin shortages [18]. Both medical healthcare workers and patients seek online information on anaphylaxis by using social media and search engines, thus may include multiple patient risks [19]. In addition, online sources may provide unreliable information on medicine shortages.

Authors' previous studies have shown that searches and article openings in the dedicated online databases could be used to detect various infectious disease epidemics in Finland. The first study found that HCPs' internet searches for Lyme borreliosis coincided with the diagnoses of Lyme borreliosis with clear seasonal variation (2017) [20], while the second study found similar seasonal variations in Lyme disease–related information-seeking behaviors between citizens and HCPs, and media coverage may have affected HCPs' and citizens' searching behavior (2017) [21]. In the third and fourth studies, HCPs' searches for oseltamivir in the dedicated databases correlated with influenza diagnoses in adults (2019) [22] and children (2022) [23], indicating that the searches could be used as a supplementary source of information when detecting influenza epidemics. The fifth study mathematically analyzed and modeled the database logs used by HCPs and citizens, thus could be used as an additional source of information for surveilling COVID-19 (2021) [24].

## Research gap

Even though studies have been conducted on medicine shortages in general, there was only a little prior knowledge of epinephrine auto-injector shortages and to what extent this has been researched. Some studies were identified regarding the unavailability of epinephrine in community spaces [10, 11], people carrying expired auto-injectors [12], or not having auto-injectors with them [11]. These results could not satisfy the demand for the study objective. Therefore, the research field was decided to be reviewed more closely.

## Literature review

A literature review was performed in order to search for the research field of epinephrine auto-injector shortages. Four databases were used: PubMed, Ovid MEDLINE, Scopus, and the Cochrane Library. The following search was run: "(medicine shortage* AND epinephrine) OR (medicine shortage* AND adrenaline) OR (drug shortage* AND epinephrine) OR (drug shortage* AND adrenaline) OR (autoinjector shortage*) OR (epinephrine shortage*) OR (adrenaline shortage*) OR (EpiPen shortage*) OR (Jext shortage*) OR (Emerade shortage*) OR (epinephrine autoinjector availability)". No time limits, language requirements, or specific study designs were set. A total of 366 articles were found in the search (212 from PubMed, 8 from Ovid MEDLINE, 123 from Scopus, and 23 from the Cochrane Library). After evaluation and duplicate removal, only 5 relevant articles [13, 25–28] on epinephrine auto-injector shortages could be found and decided to analyze.

A survey study conducted in 2005 [13] found that there are concerns about the lack of availability and affordability of epinephrine auto-injectors in out-of-hospital treatment of anaphylaxis, especially among the pediatric population. The 150µg auto-injectors for children were available in 43.6% of the 39 countries, while the 250–300µg auto-injectors for adults were available in 56.4% of the countries. The limited auto-injector availability was found in Asia, South America, and Africa, whereas the widespread availability appeared in Europe, as well as availability occurred in the USA, Canada, and Australia.

A commentary in 2016 [25] discussed epinephrine auto-injector shortages, especially considering drug prices, although regulation, manufacturing problems, and supply chain issues influence shortages. It has also been suggested that the drug market should be opened, and the epinephrine delivery device for anaphylaxis should be given expedited approval for import to the USA, since these are already available in Britain, Canada, and the European Union.

An editorial in 2018 [26] stated that the EpiPen® shortage began in November 2017, and then spread worldwide. EpiPen® has a strong market position, and only a few comparable medications exist. EpiPen® shortages have been reported in Australia, Canada, the USA, and the UK. Due to EpiPen® shortages, professionals and citizens have become more aware of the alternative auto-injectors available. This has also resulted in the reduction of alternative auto-injectors in stocks, and HCPs in several countries have been advised to prescribe alternative products that are more affordable for many patients.

An editorial in 2017 [27] stated that the availability of epinephrine auto-injectors is limited to only 32% of all the world's 195 countries that mostly include high-income countries. A review in 2020 [28] found that there are four epinephrine auto-injectors available in France (2018), namely EpiPen®, Jext®, Emerade®, and Anapen®. It has also been stated that the awareness and availability of generic epinephrine auto-injectors should be increased [27], since the costs, national regulations, and poor evidence about the value of epinephrine may affect the lack of auto-injector availability. In addition, accurate data on the epidemiology of anaphylaxis morbidity and mortality may be limited [27] and located in registers with limited scope and population sources [28]. It is important to report the data properly with validation included. Therefore, an action plan has been created to provide worldwide data on the consumption of epinephrine auto-injectors, including knowledge of prescriptions, to forecast the auto-injector market [28].

Since only five articles on epinephrine auto-injector shortages could be found in the literature review, this current study was conducted to fill the research gap. More specifically, there have been no prior studies on internet information seeking by HCPs or citizens during epinephrine shortages. Notably, an action plan [28] suggests that more data on epinephrine auto-injectors are needed. Therefore, this study brings novel knowledge into the detection of epinephrine auto-injector shortages by using the dedicated online sources of reliable medical information.

## Purpose of the study

The general aim of this study was to assess online information seeking on epinephrine auto-injectors and its relation to epinephrine shortages in Finland. The first research question is whether there is a relation between HCPs' searches for auto-injectors and epinephrine auto-injector shortages. The second research question is whether citizens' openings of anaphylaxis articles are related to shortages.

## Hypotheses

Since HCPs seek information on auto-injectors, their strengths, package sizes, and adverse effects, the hypothesis contrived that epinephrine shortages might be detectable in the user

data of the dedicated online databases for professionals. It was also hypothesized that citizens' interest in epinephrine shortages might be shown in the openings of anaphylaxis articles in the databases for citizens.

## Materials and methods

### Setting

In Finland, Physician's Database (PD) serves as an online medical source with evidence-based information aimed at HCPs working in daily practice, while Health Library (HL) provides health-related articles for citizens and professionals. Both PD and HL are produced and maintained by the Duodecim Publishing Company Ltd, and the databases are available throughout the entire healthcare system across the country. The Finnish Medical Society Duodecim [29] is Finland's largest scientific association and owns Duodecim Publishing Company Ltd publishing information content for medical and healthcare professionals. PD includes a pharmaceutical database with information on epinephrine auto-injectors. Once professionals open the page with information on epinephrine, PD stores HCPs' searches in the log files. PD is widely used by various professionals in the Finnish healthcare, such as physicians, pharmacists, and nurses. In 2022, there were approximately 22,000 working-age physicians and 8,519 pharmacists nationwide, whereas working-age nurses equaled 76,581 in 2021. Approximately 50 million searches a year are done in the PD database, whereas HL article accesses equal over 50 million a year. Citizens' article openings of anaphylaxis are stored in the HL log files. In 2022, the population of Finland was 5,540,745 inhabitants.

The distribution chain of medicines in Finland is closely controlled and strictly managed by professionals. This ensures the availability of medicines and prevents counterfeit medicines from entering the distribution system. All medicines arrive at the pharmaceutical wholesaler [30], and there are only a few wholesalers dealing with medicine distribution nationwide. The distribution of medicines is based on the one-channel principle, meaning that the pharmaceutical company's products are provided by only one wholesaler to pharmacies. There are a total of 800 pharmacies across the country. Pharmaceutical companies in Finland must inform the wholesaler about medicine shortages, and the wholesaler provides information to the national authority, Finnish Medicines Agency (Fimea) [31], regulating pharmaceuticals nationwide. Fimea has defined "medicine shortage" as a temporary unavailability of a product at the wholesaler, but there may be some products left in pharmacies. Fimea publishes the notifications of medicine shortages on their website [31], providing early information for HCPs and other pharmaceutical companies to prepare for shortages and alternative products. "Auto-injector" is defined as a medical device (self-administrable and injectable medications) that patients carry with them, such as epinephrine auto-injectors. In case of the shortage of auto-injectors, there is lack of auto-injectors in pharmacies, thus the demand for auto-injectors is not met.

The Social Insurance Institution of Finland (Kela) maintains the register of medications prescribed by HCPs [32, 33]. Kela collects the data from pharmacies delivering medication and provides statistical databases on its website. Kela has an open-access source of prescribed medications based on the Anatomical Therapeutic Chemical (ATC) codes. No personal information on prescriptions is contained in the data.

### Descriptive analysis

The literature review found one article in 2005 and four articles in 2016–2020. Therefore, the data on searches and openings were collected during January 2016–December 2022, including monthly log data on epinephrine auto-injector searches and anaphylaxis article openings across the country. In Finland, the two epinephrine auto-injectors (EpiPen® and Jext®) were

on the market during 2016–2022, but the third alternative (Emerade®) became available in June 2018, and the data were therefore not available before that. The three auto-injectors are not exchangeable with one another in pharmacies. If any medicine shortage appears, an alternative auto-injector prescription by a physician is needed. The descriptive analysis included PD database searches of the seven available auto-injectors in Finland (EpiPen Jr® 150μg, EpiPen® 300μg, Jext® 150μg, Jext® 300μg, Emerade® 150μg, Emerade® 300μg, and Emerade® 500μg) and HL database openings of the four anaphylactic-related articles ("Use of Adrenaline Pen", "Anaphylactic Reaction (Sudden Hypersensitivity Reaction)", "Bites and Stings", and "Venomous Bites and Stings"). All articles are in Finnish. Monthly log data were retrospectively collected by using anonymous Internet Protocol addresses where users could not be identified.

Fimea provided register-data on epinephrine shortages comprising the start and end of each shortage from February 2018 to December 2022. No previous data were available. In addition, the open-access web-based data were collected from Kela, comprising epinephrine quarter-year prescriptions from January 2019 to December 2022. The Anatomical Therapeutic Chemical (ATC) code "C01CA24" for epinephrine was used. Prescription data before 2019 were not available, and no prescriptions for various auto-injectors (EpiPen®, Jext®, Emerade®) and their strengths were distinguishable. Only log- and register-data were included in this study with no human subjects involved. The searches, article openings, shortage periods, and prescriptions were visualized and tabled.

## Results

Visual peaks of EpiPen® auto-injector PD searches by HCPs appeared from April to August in Finland 2016–2022 (Fig 1). Searches for other epinephrine auto-injectors (Jext® and Emerade®) did not show clear seasonal patterns across the years. However, the major peak of searches for epinephrine auto-injectors (EpiPen®, Jext®, Emerade®) appeared from June to August 2018, when the maximum number of searches for EpiPen® 300μg in July 2018 was double to what it was in other seasons.

Shortages of EpiPen® auto-injectors started in November 2017 [26] and appeared in Finland during February 2018–December 2022. The major searching peak in July 2018 occurred only a few months after the start of the EpiPen® shortage in February 2018 (Fig 1). The Jext® shortage appeared during October–November 2020, and the Emerade® shortage began in February 2020. No peaks appeared in the searches of Emerade® and Jext® after the shortages began. However, seasonal peaks in EpiPen® were observed (Fig 1).

In citizens' openings of anaphylaxis articles in HL, the "Bites and Stings"–article showed visually similar seasonal patterns as could be seen in searches of EpiPen® (Figs 1 and 2). The two major peaks in openings of "Use of Adrenaline Pen" and "Bites and Stings"–articles appeared during the summer months of 2018, only a couple of months later when EpiPen® shortages began in February 2018. "Anaphylactic Reaction (Sudden Hypersensitivity Reaction)"–article openings showed a great peak in December 2020 when the COVID-19 vaccinations started in Finland (Fig 2). The two-month-long Jext® shortage hardly explains the following peak in the article opening.

The number of epinephrine auto-injector searches in PD and anaphylaxis article openings in HL is shown in Table 1. During the major EpiPen® shortage in 2018, PD searches increased by 54% (14,207 to 21,843) in EpiPen® and 104% (3,222 to 6,557) in Jext® from 2017 to 2018. The searches remained at a lower level in 2019–2022. Emerade® searches in 2018 decreased by 66% (10,248 to 3,527) in the next year of 2019, and no comparable PD data were available until June 2018.

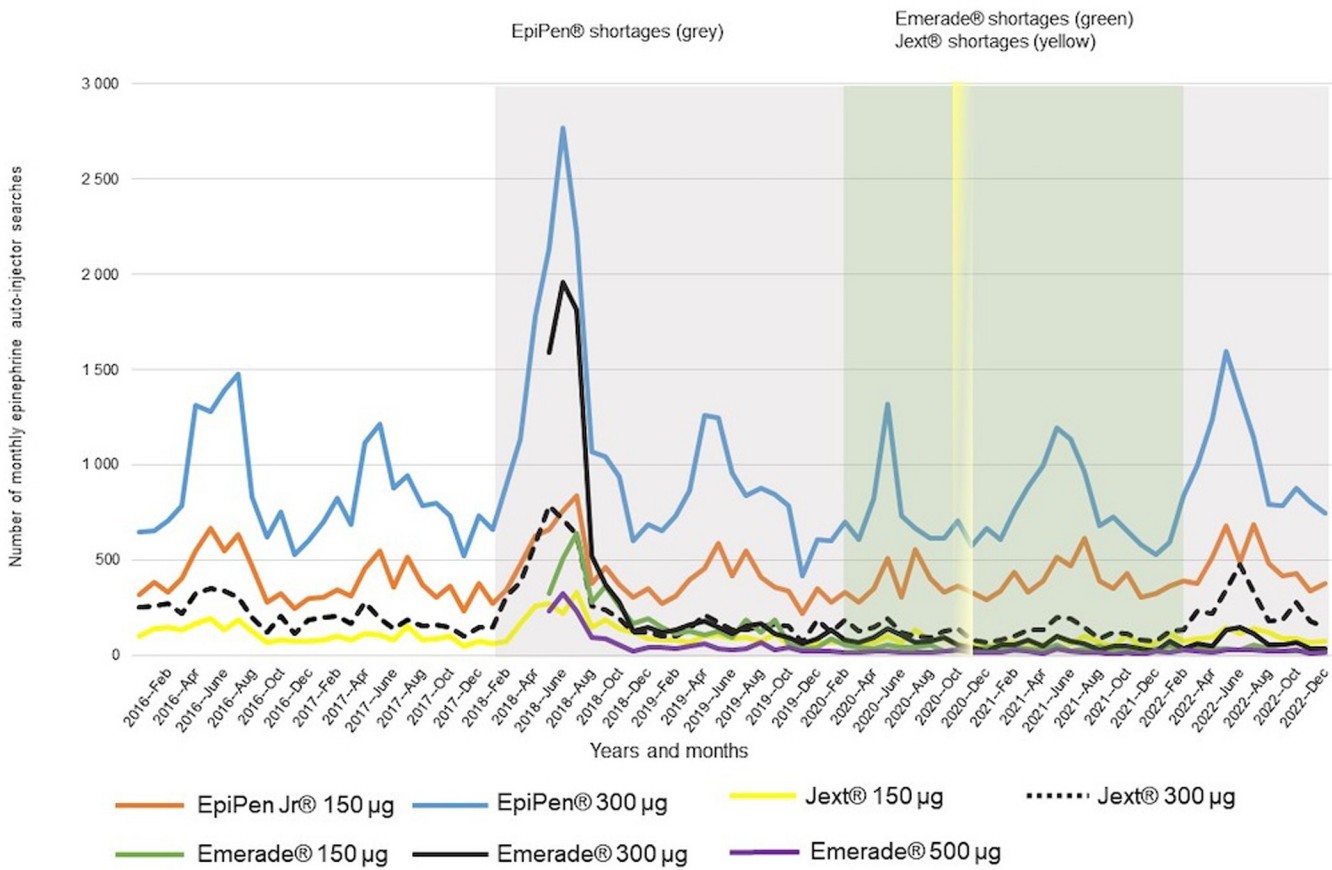

**Fig 1. Physician's Database searches of epinephrine auto-injectors in Finland 2016–2022, as well as epinephrine shortage periods of EpiPen® (Feb 2018–Dec 2022, grey), Jext® (Oct–Nov 2020, yellow), and Emerade® (Feb 2020–Feb 2022, green).**

Shortages in different strength and package sizes of auto-injectors were reported by Fimea. The start and end of shortage periods were assessed on a monthly basis. Fifty EpiPen®, one Jext®, and twelve Emerade® shortage periods appeared in Finland during 2018–2022 (Table 2). Some periods that lasted less than a month are shown as the same month. There were two or three different reports during the months listed as the same starts and ends in the table. Shortages also overlapped the years. Each cell (grey, yellow, green) in the table indicates one shortage period.

Auto-injector shortage periods with starts and ends are shown in Fig 3. Since the periods may overlap years and months, visualization of shortages only corresponds to years and months (x-axis) in duration. Seasonal peaks in auto-injector prescriptions were found in 2019–2022 (Fig 3). Prescriptions almost doubled from January–March to April–June in 2019 (6,523 to 11,077), 2020 (6,599 to 10,402), 2021 (5,535 to 9,356), and 2022 (6,028 to 11,624). In July–September, auto-injectors were prescribed lesser, decreasing in October–December to the same level than prescriptions in January–March (Table 3). Peaks appeared during summertime, similarly to EpiPen® searches and "Bites and Stings"–article openings (Figs 1–3). Lower peaks and deeper troughs in prescriptions during 2020–2022 were visually related to the periods of all three auto-injector (EpiPen®, Jext®, Emerade®) shortages. Interestingly, the great peak in article openings of "Anaphylactic Reaction" appeared during November 2020–February 2021 when auto-injector prescriptions were lowest and COVID-19 vaccinations started (Figs 2 and 3).

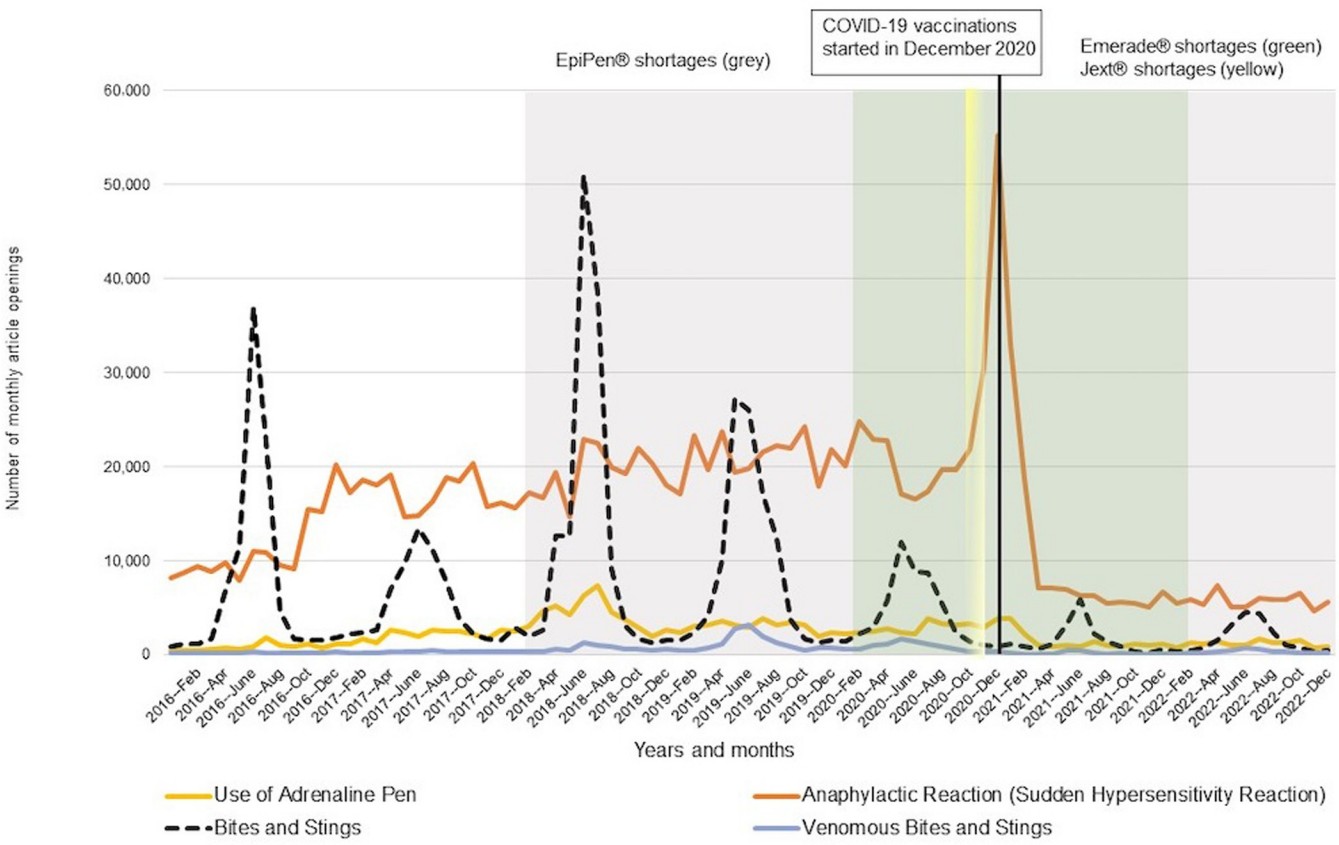

**Fig 2. Health Library article openings of "Use of Adrenaline Pen", "Anaphylactic Reaction (Sudden Hypersensitivity Reaction)", "Bites and Stings", and "Venomous Bites and Stings" in Finland 2016–2022, as well as epinephrine shortage periods of EpiPen®, Jext®, and Emerade®, with the start of COVID-19 vaccinations.**

**Table 1. Physician's Database epinephrine auto-injector searches and Health Library anaphylaxis article openings in Finland 2016–2022.**

| | Year | | | | | | | |
|---|---|---|---|---|---|---|---|---|
| **Number of auto-injector searches in Physician's Databases** | **2016** | **2017** | **2018** | **2019** | **2020** | **2021** | **2022** | **2016–2022** |
| **EpiPen Jr® 150µg** | 5,156 | 4,400 | 5,870[b] | 4,662 | 4,389 | 4,855 | 5,497 | 34,829 |
| **EpiPen® 300µg** | 10,985 | 9,807 | 15,973 | 10,161 | 8,568 | 9,850 | 11,567 | 76,911 |
| **Jext® 150µg** | 1,545 | 1,102 | 2,047 | 992 | 904[b] | 810 | 1,195 | 8,595 |
| **Jext® 300µg** | 2,955 | 2,120 | 4,510 | 1,650 | 1,610 | 1,440 | 2,751 | 17,036 |
| **Emerade® 150µg** | N/A[a] | N/A | 2,536 | 1,487 | 527[b] | 417 | 393 | 5,360 |
| **Emerade® 300µg** | N/A | N/A | 6,658 | 1,567 | 1,037 | 640 | 848 | 10,750 |
| **Emerade® 500µg** | N/A | N/A | 1,054 | 473 | 219 | 203 | 270 | 2,219 |
| **Number of article openings in Health Library** | | | | | | | | |
| **Use of Adrenaline Pen** | 9,948 | 23,658 | 49,079 | 36,708 | 33,124 | 19,386 | 14,388 | 186,291 |
| **Anaphylactic Reaction** | 124,150 | 212,821 | 226,901 | 248,999 | 255,312 | 162,325 | 69,884 | 1,300,392 |
| **Bites and Stings** | 92,334 | 65,811 | 139,288 | 109,190 | 53,564 | 18,846 | 20,273 | 499,306 |
| **Venomous Bites and Stings** | 2,556 | 3,867 | 7,191 | 14,737 | 10,390 | 2,411 | 3,714 | 44,866 |

[a]N/A: Data not available until June 2018

[b]Shortage years of EpiPen® (grey), Jext® (yellow), and Emerade® (green) auto-injectors

**Table 2. Starts and ends of monthly epinephrine auto-injector shortages (Finnish Medicines Agency) in Finland 2018–2022.**

| Auto-injector | Year | | | | | | | | | | Number of reported shortages[d] |
| | 2018 | | 2019 | | 2020 | | 2021 | | 2022 | | |
| | Start | End | Start | End | Start | End | Start | End | Start | End | |
|---|---|---|---|---|---|---|---|---|---|---|---|
| **EpiPen®**[a] | Feb | | | Jan | Jan | Mar | Feb | Feb | Jan | Feb | 50 |
| | | | Mar | Apr | Jan | Apr | Feb | Feb | Jan | Feb | |
| | | | July | Sep | Mar | Apr | Feb | Feb | Feb | Feb | |
| | | | Aug | Nov | Apr | May | Feb | Mar | July | July | |
| | | | Oct | Nov | | | Apr | May | July | Aug | |
| | | | Nov | | Mar | | May | May | Aug | Sep | |
| | | | Dec | | Mar | | June | July | Sep | Oct | |
| | | | | | | | June | July | Sep | Nov | |
| | | | | | Apr | May | Aug | Oct | Oct | Nov | |
| | | | | | May | May | Aug | Nov | Nov | Dec | |
| | | | | | May | Aug | Aug | Dec | | | |
| | | | | | June | July | Sep | Sep | | | |
| | | | | | July | Sep | Sep | Oct | | | |
| | | | | | Aug | Aug | Sep | Nov | | | |
| | | | | | Aug | Sep | Oct | Nov | | | |
| | | | | | Aug | Oct | Dec | Dec | | | |
| | | | | | Sep | Sep | | | | | |
| | | | | | Sep | Oct | | | | | |
| | | | | | Sep | Dec | | | | | |
| | | | | | Nov | Nov | | | | | |
| | | | | | Nov | Dec | | | | | |
| **Jext®**[b] | | | | | Oct | Nov | | | | | 1 |
| **Emerade®**[c] | | | | | Feb | Dec | Jan | Feb | | | 12 |
| | | | | | | | Jan | Feb | | | |
| | | | | | | | Feb | May | | | |
| | | | | | | | June | Sep | | | |
| | | | | | | | Sep | Oct | | | |
| | | | | | | | Oct | Nov | | | |
| | | | | | | | Oct | Nov | | | |
| | | | | | | | Nov | Dec | | | |
| | | | | | | | Nov | Dec | | | |
| | | | | | | | Nov | | | Feb | |
| | | | | | | | Dec | | | Feb | |

[a]EpiPen® shortages in 2018–2022 (grey)

[b]Jext® shortages in 2020 (yellow)

[c]Emerade® shortages in 2020–2022 (green)

[d]Includes the start and end of a shortage period in each cell.

## Discussion

### Principal findings

This study could characterize epinephrine shortages and their temporal relation to HCPs' and citizens' online information seeking in the dedicated medical databases in Finland 2016–2022. Visually similar seasonal patterns of auto-injector searches and anaphylaxis article openings

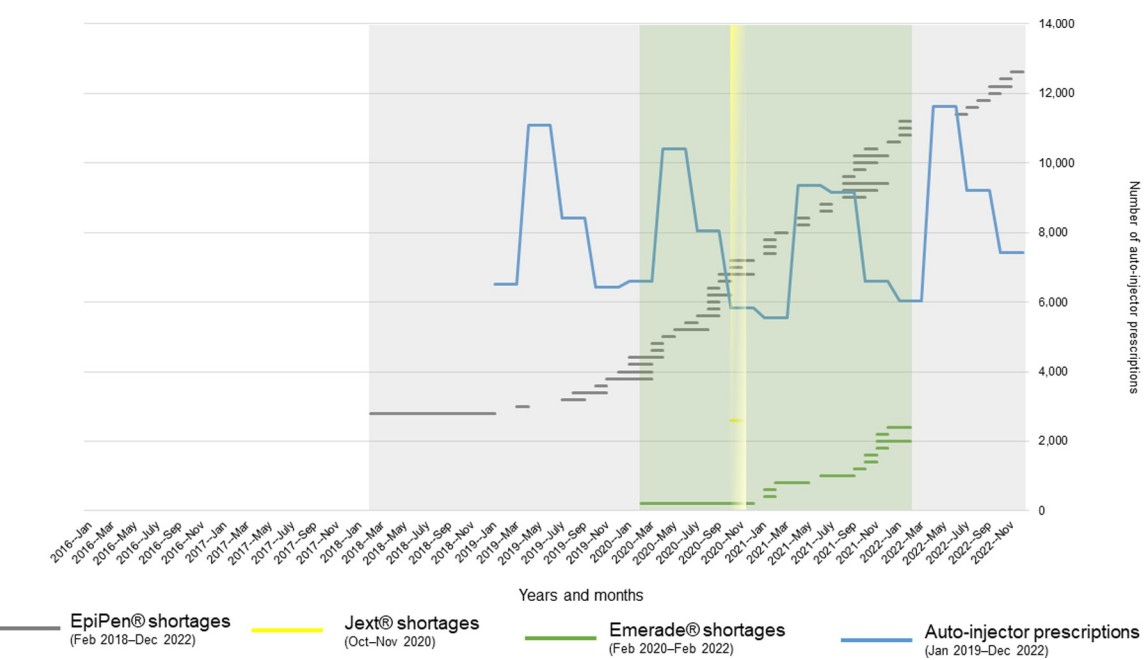

**Fig 3. Auto-injector shortage periods of EpiPen®️ (grey), Jext®️ (yellow), and Emerade®️ (green) by Finnish Medicines Agency, and auto-injector prescriptions by Social Insurance Institution (blue).** No data available during January 2016–2018.

(Figs 1 and 2), as well as auto-injector prescriptions (Fig 3), were identified during shortages. Prescriptions decreased during the periods of three simultaneous auto-injector shortages.

## Interpretation of the results

The results of PD searches and HL article openings of interest were visually found around the summer months in 2018 when EpiPen®️ shortages prevailed in Finland [14] (Fig 3 and Table 2) and worldwide [26, 28]. The maximum search or openings peaks appeared during this period, including auto-injector searches of EpiPen Jr®️ 150μg (August), EpiPen®️ 300μg (July), Jext®️ 150μg (August), and Jext®️ 300μg (June), as well as anaphylaxis articles of "Use of Adrenaline Pen" (August) and "Bites and Stings" (July). The maximum searches of Emerade®️ compared to previous seasons could not be analyzed, since no search data existed until June 2018 (Fig 1 and Table 1). However, Emerade®️ 300μg searches showed a great peak in July 2018, possibly caused by a sales license for Emerade®️ issued by Fimea, to mitigate the long-lasting epinephrine auto-injector shortage. This information on a sales license has also been published on professional websites, thus may have increased HCPs' knowledge of further information seeking on auto-injectors. HCPs' searches for auto-injectors and epinephrine

**Table 3. Auto-injector quarter-year prescriptions (Social Insurance Institution) in Finland 2019–2022.**

| | Year and months | | | | Year and months | | | |
|---|---|---|---|---|---|---|---|---|
| | **2019** | | | | **2020** | | | |
| | **Jan–Mar** | **Apr–June** | **July–Sep** | **Oct–Dec** | **Jan–Mar** | **Apr–June** | **July–Sep** | **Oct–Dec** |
| **Number of prescriptions** | 6,523 | 11,077 | 8,401 | 6,417 | 6,599 | 10,402 | 8,052 | 5,834 |
| | **2021** | | | | **2022** | | | |
| | **Jan–Mar** | **Apr–June** | **July–Sep** | **Oct–Dec** | **Jan–Mar** | **Apr–June** | **July–Sep** | **Oct–Dec** |
| **Number of prescriptions** | 5,535 | 9,356 | 9,140 | 6,609 | 6,028 | 11,624 | 9,208 | 7,416 |

auto-injector shortages were shown to relate to each other, indicating that professionals seek online information during shortages, especially at the beginning of the shortage in 2018. Media coverage on various internet platforms and social media may increasingly have affected professionals' interest in information seeking as well.

The study findings indicate that HCPs and citizens seek online information on auto-injectors and anaphylaxis during the shortages of epinephrine auto-injectors. Although shortages in Finland started in February 2018, searches and openings peaked later in the summer months. This is probably caused by the cumulative shortage of auto-injectors in pharmacies, people moving outdoors during summertime encountering insects, and the increased knowledge of shortages among HCPs and citizens (e.g., media coverage). This may explain the summer peaks in searches and openings. Prior literature has shown that citizens carry expired auto-injectors [12], thus may appear in peaks since citizens seek more online information on epinephrine.

The longer and wider shortages may result in harmful workload for HCPs [8] and increased expenses [5], although most shortages can be easily resolved in pharmacies by choosing an alternative product. A previous study has shown that medicine shortages have mostly not caused any problems in pharmacies, and the customers have received their medicine because of the availability of a substitutable product [8]. This current study supports these findings (Fig 1 and Table 1). As shown in Table 1, EpiPen® searches increased by 54% and Jext® searches by 104% in 2017–2018. Emerade® searches decreased by 66% from 2018 to 2019. It is concluded that HCPs seek information on auto-injectors during EpiPen® shortages in 2018, but also on auto-injectors available (Jext®, Emerade®).

Citizens' openings of anaphylaxis articles showed visual relation to summer months during 2016–2022 (Fig 2). The highest peak in "Bites and Stings" –article openings appeared in June 2018, then decreasing every summer once the medicine shortages continued. This may be explained by citizens' tiredness of overwhelming information on auto-injector shortages published in online sources or in the mass media. An unexpected peak of citizens' article openings of "Anaphylactic Reaction" occurred at the end of 2020 (Fig 2) when COVID-19 vaccinations started in Finland. Citizens may have searched for post-vaccination symptoms, such as allergic reactions, in the HL database. This finding supports the previous results on citizens' seeking behavior in the dedicated internet database [21, 24]. Searches slightly increased during the shortages of EpiPen®, Jext®, and Emerade® in February 2018–December 2020, meaning that mass media may have provided health information on shortages resulting in citizens' online information seeking in the HL database. However, searches suddenly dropped after the peak continued at a lower lever even though vaccinations proceeded. This can be explained by citizens' searching behavior, including tiredness of continuous news on COVID-19 in the mass media, or citizens searching for health information in other sources. The article "Bites and Stings" showed an over 11-fold (499,306/44,866) number of searches compared to the article "Venomous Bites and Stings" (Table 1). It is likely that citizens type the search word "Bites" or "Stings" in general search engines, thus clicking the article "Bites and Stings" first and finding the HL website. This may have resulted in a decrease in openings in the latter article. Insect bites and stings, caused by wasps, bees, hornets, or mosquitos, are not typically regarded as venomous, although they may cause serious allergic reactions (anaphylaxis) in adults and children [9]. In addition, "Bites and Stings" visually appeared to have clear seasonal patterns compared to "Venomous Bites and Stings" (Fig 2).

Lesser auto-injector prescriptions appeared during the periods of three simultaneous auto-injector shortages (EpiPen®, Jext®, Emerade®) in October 2020–March 2021 (Fig 3). This means that prescriptions and shortages visually coincided with each other. At the same time, citizens' article openings of "Anaphylactic Reaction" showed a great peak, possibly meaning

that media coverage on the start of COVID-19 vaccinations has affected online seeking behavior among the citizens. This has probably resulted in an increase in the openings of anaphylaxis articles. Since the "Anaphylactic Reaction" –article showed the greatest number of openings in the HL database during 2016–2022, citizens may have found the HL website due to one of the first results provided in the general search engines. More specifically, this has occurred during the shortages when citizens' thirst for knowledge of auto-injectors might have been greatest. Media coverage, online information, and social media may have affected HCPs' and citizens' information seeking in PD and HL. In addition, the combination of overwhelming auto-injector information on these platforms might have led to an increase in searches and article openings in the databases.

In Finland, community pharmacies delivering medicine in the sparsely-populated country with long distances, may encounter challenges in medicine shortages [7, 8]. Since the medication is not substitutable for another, overcoming shortages requires cooperation with pharmacies and physicians, and occasionally with medical authorities as well. The situations and solutions are usually unique. This current study brings novel knowledge into the field of medicine shortages, and the prior work [20–24] supports the findings that HCPs and citizens seek real-time information in the dedicated medical online databases. This information on searching auto-injectors on the internet can be used in detection of epinephrine shortages.

## Strengths and limitations

The strength of this study was the representativeness of log data from PD and HL, as demonstrated in the previous studies of infectious diseases [20–24]. However, this study has several limitations. Some professionals may have searched for information in paper sources or consulted colleagues, while others may be familiar with auto-injectors, thus not needing additional information from PD. This phenomenon could have decreased the number of searches in PD. Not all searches are done by a prescribing physician since it is not distinguishable which HCP is performing the searches. Also, the demographics of professionals performing searches cannot be classified into groups of age, sex, or education. In this study, visual relation between searches and shortages could only be explored, meaning that similar temporal patterns were detected in the figures. Various healthcare sectors (public primary/specialized care, private care, pharmacies) are not distinguishable in the PD data. Notably, students and less-experienced junior professionals may have sought medical information during summertime, since more-experienced senior professionals are on vacation, indicating an increased number of searches during the summer. In Finnish public healthcare centers, nurses may have checked the correct auto-injector before sending the prescription request to a physician. Pharmacists may also seek auto-injector information in PD. In addition, it is important to notice that media coverage on the shortages of auto-injectors, specifically EpiPen®, may have affected both HCPs' and citizens' seeking behavior in PD and HL. This may have increased the number of searches, but media coverage and its potential effect on information seeking were not measured. Seasonal search patterns during summertime might have some influence on increased information seeking because of more leisure time among HCPs and citizens. Some patients have possibly requested a professional to prescribe an auto-injector, hence showing the increased number of searches in PD. Prescription data on auto-injectors (i.e., ATC code) may include epinephrine ampules, but it is presumed that physicians almost singularly prescribe injectors for patients needing epinephrine outside healthcare. The data were not available for some years. Searches for EpiPen® and Jext® could not be compared with Emerade® due to the unavailable data on Emerade® auto-injectors during January 2016–May 2018. Also, the data on EpiPen® shortages before 2018 and auto-injector prescriptions before 2019 were not available, thus no comparison could

be made. The literature review was performed in order to fill the research gap, yielding only five relevant articles for further analysis. This means that little prior information on epinephrine shortages [13, 25–28], exists in the scientific literature and authorities' reports. An action plan [28] suggested that further research on epinephrine is needed.

## Conclusion

Since only a little prior research on epinephrine auto-injector shortages existed, this study was conducted by using online information on HCPs' searches of auto-injectors and citizens' openings of anaphylaxis articles to detect epinephrine shortages. To authors' knowledge, this is the first study to analyze the topic from this point of research methodology, since there were no prior studies on internet seeking by professionals and citizens during epinephrine shortages. This research achieved the general aim by showing the relation between online information seeking on epinephrine auto-injectors and epinephrine shortages in Finland. Two research questions could be answered: Epinephrine auto-injector shortages were related to HCPs' searches for auto-injectors, and citizens' openings of anaphylaxis articles related to summer months and shortages with lesser prescriptions. The hypotheses were also answered: This study could show that epinephrine shortages were detectable in the HCPs' user data of the dedicated online databases, and citizens' interest in epinephrine shortages was shown in anaphylaxis article openings. Therefore, it is concluded that the dedicated internet databases aimed at professionals and citizens could be used as an additional source of information to detect anaphylactic reactions and epinephrine auto-injector shortages.

The conclusions of the study implicate that databases should be combined, information shared, and medicine shortages prepared for, as well as cooperation strengthened. First, various healthcare databases should be merged into one common database that provides easy-to-access usage for professionals working in public (primary/specialized) and private healthcare as well as in administrative duties and research. Notably, artificial intelligence [34] should be embedded in systems that analyze the data on medicine shortages in real-time to detect shortages. In the future healthcare, artificial intelligence, including machine learning, deep learning, algorithms, and robots, will increasingly prevail. Large volumes of log- and register-data can effectively be analyzed, thus enhancing data processing and interpretation. This improves professionals' daily work in diagnosing and treating patients.

Second, healthcare units (i.e., healthcare centers, hospitals, pharmacies) and national authorities should easily share early information on upcoming medicine shortages to prepare for alternative products. This is important during summertime since senior professionals may be on vacation and juniors with less experience work in the summer. As shown in this study, epinephrine shortages and information seeking certainly occur in summer months. HCPs' searches in the databases remained at a similar level even though epinephrine shortages continued. This may cause a burdensome workload for professionals if the shortages last for years and alternative products are not available. Well-being at work is crucial among professionals during shortages.

Third, updating training on shortages for various professionals and managers should be developed by providing additional courses on patient safety, leadership, decision-making, and evidence-based medicine. Continuity in medicine distribution strengthens healthcare quality. Transparency and cooperation between multidisciplinary organizations are also essential, not only regionally and nationally but also worldwide. Awareness of anaphylaxis and auto-injectors should be increased among professionals and citizens, and the availability of affordable auto-injectors on the market is essential as well. Pharmaceutical companies must be involved in cooperation.

Finally, novel research methods and materials should be developed when conducting studies on medicine shortages, including online information sources of register- and log-data. It is important to empower both HCPs and citizens by taking part in the improvement of databases on the internet and being involved in the development of reliable medical and health information. The users of the dedicated databases in countries with different healthcare systems need to be characterized in detail to analyze online information seeking behavior in preparing for medicine shortages in the future.

## Supporting information

**S1 Data.**
(XLSX)

## Acknowledgments

The authors would like to express their gratitude to Johanna Linnolahti who provided the register-data from the Finnish Medicines Agency (Fimea).

## Author Contributions

**Data curation:** Milla Mukka.

**Formal analysis:** Milla Mukka, Samuli Pesälä.

**Resources:** Pekka Mustonen.

**Supervision:** Samuli Pesälä, Minna Kaila, Otto Helve.

**Writing – original draft:** Milla Mukka.

**Writing – review & editing:** Samuli Pesälä.

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
