## [Decision Letter · Decision Letter 0]

11 Jul 2023

PONE-D-23-19109Detecting epinephrine auto-injector shortages in Finland 2016‒2022: Log-data analysis of online information seekingPLOS ONE

Dear Dr. Mukka,

Thank you for submitting your manuscript to PLOS ONE. After careful consideration, we feel that it has merit but does not fully meet PLOS ONE’s publication criteria as it currently stands. Therefore, we invite you to submit a revised version of the manuscript that addresses the points raised during the review process.

 Please submit your revised manuscript by Aug 25 2023 11:59PM. If you will need more time than this to complete your revisions, please reply to this message or contact the journal office at plosone@plos.org. Please include the following items when submitting your revised manuscript:A rebuttal letter that responds to each point raised by the academic editor and reviewer(s). You should upload this letter as a separate file labeled 'Response to Reviewers'.A marked-up copy of your manuscript that highlights changes made to the original version. You should upload this as a separate file labeled 'Revised Manuscript with Track Changes'.An unmarked version of your revised paper without tracked changes. You should upload this as a separate file labeled 'Manuscript'.

We look forward to receiving your revised manuscript.

Kind regards,

Radoslaw Wolniak, full professor

Academic Editor

PLOS ONE

Journal Requirements:

   "NO. This is a unfunded study."

Additional Editor Comments:

Please adjust the paper accoring to reviewers comments.. 

Reviewers' comments:

Reviewer's Responses to Questions

**Comments to the Author**

1. Is the manuscript technically sound, and do the data support the conclusions?

Reviewer #1: Partly

Reviewer #2: Partly

2. Has the statistical analysis been performed appropriately and rigorously? 

Reviewer #1: Yes

Reviewer #2: I Don't Know

3. Have the authors made all data underlying the findings in their manuscript fully available?

Reviewer #1: Yes

Reviewer #2: Yes

4. Is the manuscript presented in an intelligible fashion and written in standard English?

Reviewer #1: Yes

Reviewer #2: Yes

5. Review Comments to the Author

Reviewer #1: Dear Authors,

Congratulations for your interesting research. I have some suggestions on how to make your text more attractive

In the introduction, you write about the previous research. I suggest that you briefly explain what the research was, when it was conducted, perhaps refer to publications if published.

In the introduction, please state the originality of the material presented. I suggest in the introduction specify the methodology of the research. The diagnosis itself should indicate the novelty of the results and the publication of the considerations in scientific journals. The current state of the research field should be thoroughly reviewed and key publications should be cited.

Some amendments in the format are due to admit publication. The main problem is the epistemological structure (why the article was conceived and how the study was developed). I suggest the following structure of objectives: (i) research gap; (ii) purpose of the article; (v) assumptions or hypo; and (vi) research method. This structure must appear in the introduction.

The research gap must be created by a systematic literature review that provides 'holes' in the state of knowledge on the topic. I believe that a full review should not be done, but an analysis of about 5-8 studies on the topic under discussion. You can find some examples, which will show the relevance of the issue, as it is indeed a topic of current, relevant research. At the end of the justification you should write something like: According to what we were able to find, there are no studies referring and reporting on ... With this you have therefore proven that the issue is relevant, and you have also proven that your study does indeed fill a research gap.

In the discussion, I suggest that describe the significance of the research and its impact on the wider field, show how the information obtained can be further used

Conclusions must be clearly and unambiguously linked to the results of the study. Their theoretical and practical implications should be indicated

The methodology should indicate the time period that was considered in collecting data on drug shortages.

Good luck!

Reviewer #2: PLOS One review 10 July 2023

Dear Authors,

The paper is interesting. Drug shortage is an important issue. The analysis is made of improvement. As a reviewer, I have a few comments:

Note 1: Keywords: In my opinion there are too many keywords.

Note 2: Introduction: several references for one sentence in not good: an example (60-61): Medicine shortages may cause harm and even life-threatening situations for patients encountering medication errors and adverse events [1,2,4-6]. In my opinion, authors have to write more about the events… along for each reference.

Reference in sentence (62) In different countries, the reasons for 63 medicine shortages may vary [4,7]…

Many single sentences have their own references, this is not good. Authors need to write more about what each reference contributes to their research topic.

Aim (97 etc) there are two aims in one. I propose to prepare separate aim in the form; General aim:

And RQ: research questions: RQ1:… RQ2:…..

Note 3: Setting. In my opinion is OK but I think there should be more information about the situation in the country regarding the drug distribution system, maybe what scheme....

Note 4: analysis without comments

Note 5. Conclusion , too short. Describe the aims achieved, show why analysis was done.

Best wishes

Reviewer

6. PLOS authors have the option to publish the peer review history of their article (what does this mean?). If published, this will include your full peer review and any attached files.

Reviewer #1: No

Reviewer #2: **Yes: **Bożena Gajdzik

---

## [Author Response · Author response to Decision Letter 0]

13 Aug 2023

Dear Prof. Radoslaw Wolniak and Reviewers #1 and #2, 

Thank you for your excellent suggestions in order to improve our manuscript. We have now provided the revisions described below.

Kind regards,

Dr. Milla Mukka

Response to Reviewers 

Reviewer #1: Dear Authors,

Congratulations for your interesting research. I have some suggestions on how to make your text more attractive

Response: Thank you for your comment.

In the introduction, you write about the previous research. I suggest that you briefly explain what the research was, when it was conducted, perhaps refer to publications if published.

Response: Thank you for your excellent comment. We have now added more information on our previous studies by providing details, years, and publications [20-24].

In the introduction, please state the originality of the material presented. I suggest in the introduction specify the methodology of the research. The diagnosis itself should indicate the novelty of the results and the publication of the considerations in scientific journals. The current state of the research field should be thoroughly reviewed and key publications should be cited.

Response: We have now revised Introduction by adding subsections and also the methodology of the research. We have searched and reviewed the key publications of the research field.

Some amendments in the format are due to admit publication. The main problem is the epistemological structure (why the article was conceived and how the study was developed). I suggest the following structure of objectives: (i) research gap; (ii) purpose of the article; (v) assumptions or hypo; and (vi) research method. This structure must appear in the introduction.

The research gap must be created by a systematic literature review that provides 'holes' in the state of knowledge on the topic. I believe that a full review should not be done, but an analysis of about 5-8 studies on the topic under discussion. You can find some examples, which will show the relevance of the issue, as it is indeed a topic of current, relevant research. At the end of the justification you should write something like: According to what we were able to find, there are no studies referring and reporting on ... With this you have therefore proven that the issue is relevant, and you have also proven that your study does indeed fill a research gap.

Response: We thank the reviewer for this suggestion. We have now revised Introduction by using the structure (subsections), namely “Medicine shortages”, “Auto-injectors”, “Online information seeking”, as well as the suggested subsections “Research gap”, “Purpose of the study, and “Hypotheses”. We used the subsection of “Literature search” instead of “Research method” where we describe the searching method of literature (methodology). We performed the literature search from the four databases in order to find the published articles on epinephrine auto-injector shortages. The key publications found were reviewed and analyzed. At the end of Introduction, we justify our study as suggested.

In the discussion, I suggest that describe the significance of the research and its impact on the wider field, show how the information obtained can be further used

Conclusions must be clearly and unambiguously linked to the results of the study. Their theoretical and practical implications should be indicated

Response: Thank you for your comment. We have added information on the meaning of our research by describing how our study results can be used in detection of epinephrine shortages. We have now revised the conclusion section and added information in the discussion. We have also described the achievements of the general and two aims of our study, as well as shown the reason why the analysis was done (Reviewer #2 comment). In Conclusion, we have separated this section in two, in order to show the theoretical and practical implications on the wider field as well as showing the future aspects.

The methodology should indicate the time period that was considered in collecting data on drug shortages.

Response: We performed the literature search in order to review the research field of epinephrine auto-injector shortages, yielding four key publications during 2016-2020, and one in 2005. Thus, the data were collected in our study during 2016-2022. We have mentioned this in Introduction.

Good luck!

Reviewer #2: PLOS One review 10 July 2023

Dear Authors,

The paper is interesting. Drug shortage is an important issue. The analysis is made of improvement. As a reviewer, I have a few comments:

Response: Thank you for your comment.

Note 1: Keywords: In my opinion there are too many keywords.

Response 1: Thank you for your opinion. We have limited the number of keywords. 

Note 2: Introduction: several references for one sentence in not good: an example (60-61): Medicine shortages may cause harm and even life-threatening situations for patients encountering medication errors and adverse events [1,2,4-6]. In my opinion, authors have to write more about the events… along for each reference.

Reference in sentence (62) In different countries, the reasons for 63 medicine shortages may vary [4,7]…

Many single sentences have their own references, this is not good. Authors need to write more about what each reference contributes to their research topic.

Response 2: Thank you for your comment. We have now written more text with less references embedded in one sentence as suggested. Literature is introduced more with longer sentences forming two separate paragraphs (“Medicine shortages”).

Aim (97 etc) there are two aims in one. I propose to prepare separate aim in the form; General aim:

And RQ: research questions: RQ1:… RQ2:…..

Response: Thank you for your excellent proposal. We have now clarified the aim and research questions in text. This revision is placed under the title of “Purpose of the study” in Introduction. The separation of study aims is also revised in Abstract.

Note 3: Setting. In my opinion is OK but I think there should be more information about the situation in the country regarding the drug distribution system, maybe what scheme....

Response 3: We have clarified the medicine distribution system in Finland in the setting. 

Note 4: analysis without comments

Response 4: We have added more discussion/comments on our results/analysis in the discussion section. Also, we have added the subsection of “Interpretation of results” in order to discuss and comment our study results.

Note 5. Conclusion, too short. Describe the aims achieved, show why analysis was done.

Response 5: Thank you for your comment. We have now revised the conclusion section. We have also described the achievements of the general and two aims of our study, as well as shown the reason why the analysis was done.

---

## [Decision Letter · Decision Letter 1]

22 Sep 2023

PONE-D-23-19109R1Detecting epinephrine auto-injector shortages in Finland 2016‒2022: Log-data analysis of online information seekingPLOS ONE

Dear Dr. Mukka,

Thank you for submitting your manuscript to PLOS ONE. After careful consideration, we feel that it has merit but does not fully meet PLOS ONE’s publication criteria as it currently stands. Therefore, we invite you to submit a revised version of the manuscript that addresses the points raised during the review process.

Adjust  the paper according reviewers comments. . 

We look forward to receiving your revised manuscript.

Kind regards,

Radoslaw Wolniak, full professor

Academic Editor

PLOS ONE

Journal Requirements:

Reviewers' comments:

Reviewer's Responses to Questions

**Comments to the Author**

1. If the authors have adequately addressed your comments raised in a previous round of review and you feel that this manuscript is now acceptable for publication, you may indicate that here to bypass the “Comments to the Author” section, enter your conflict of interest statement in the “Confidential to Editor” section, and submit your "Accept" recommendation.

Reviewer #2: (No Response)

Reviewer #3: (No Response)

2. Is the manuscript technically sound, and do the data support the conclusions?

Reviewer #2: (No Response)

Reviewer #3: Yes

3. Has the statistical analysis been performed appropriately and rigorously? 

Reviewer #2: (No Response)

Reviewer #3: (No Response)

4. Have the authors made all data underlying the findings in their manuscript fully available?

Reviewer #2: (No Response)

Reviewer #3: (No Response)

5. Is the manuscript presented in an intelligible fashion and written in standard English?

Reviewer #2: (No Response)

Reviewer #3: Yes

6. Review Comments to the Author

Reviewer #2: Dear Authors,

The subject is narrow, but we flow. Your analysis is good, but I have comments:

Note 1. Why there is no Literature review section. Such section should be before Materials and methods .

Note 2. Section: Setting is too short.

Note 3: Record numbers: now is 34829, should be: 34,829

Note 4: Fig 1-3 Blurred, fuzzy fonts

Note 4. Conclusion too short. Please write about reserach limitation and describle problem persented in your research.

Best wishes

Reviewer

15.09.2023

Reviewer #3: (No Response)

7. PLOS authors have the option to publish the peer review history of their article (what does this mean?). If published, this will include your full peer review and any attached files.

Reviewer #2: No

Reviewer #3: **Yes: **Dominika Marciniak

---

## [Author Response · Author response to Decision Letter 1]

17 Oct 2023

Dear Prof. Radoslaw Wolniak and Reviewer #2, 

Thank you for your suggestions in order to improve our manuscript. We have now revised our paper by providing the answers below. Grammar has also been revised throughout.

Kind regards,

Dr. Milla Mukka

Response to the Editor and Reviewer #2 

Journal Requirements: Please review your reference list to ensure that it is complete and correct.

Response: Thank you for your reminder to reference list. We have checked and updated the reference list on pages 22-26. 

Reviewer #2: 

The subject is narrow, but we flow. Your analysis is good, but I have comments:

Response: Thank you for your comment.

Note 1. Why there is no Literature review section. Such section should be before Materials 

and methods.

Response 1: Thank you for your comment. We have done the literature search and added it before Materials and Methods (on pages 6-7). The section title “Literature search” is now changed into “Literature review”. 

Note 2. Section: Setting is too short.

Response 2: Thank you for your notice. We have made this section longer by adding more information on Duodecim Publishing Company Ltd, the number of healthcare professionals in Finland, and Kela. Now, the setting section comprises three paragraphs on pages 8-9. 

Note 3: Record numbers: now is 34829, should be: 34,829

Response 3: Very good notice. We have made changes in the record numbers in the tables. 

Note 4: Fig 1-3 Blurred, fuzzy fonts

Response 4: Thank you for your comment. We have now checked all the figures and uploaded them in the PACE system. PACE accepted the figure files in resolution based on the PLosOne requirements.

Note 5. Conclusion too short. Please write about reserach limitation and describle problem persented in your research.

Response 5: Thank you for your notice. We have discussed the research limitations in its own section “Strengths and limitations” on pages 17-18. More information on the limitations and possible problems of our study has been added. New information has also been added in Conclusion on pages 19-20. Now, the conclusion section comprises five paragraphs.

---

## [Decision Letter · Decision Letter 2]

26 Dec 2023

PONE-D-23-19109R2Detecting epinephrine auto-injector shortages in Finland 2016‒2022: Log-data analysis of online information seekingPLOS ONE

Dear Dr. Mukka,

Thank you for submitting your manuscript to PLOS ONE. After careful consideration, we feel that it has merit but does not fully meet PLOS ONE’s publication criteria as it currently stands. Therefore, we invite you to submit a revised version of the manuscript that addresses the points raised during the review process.

In the Methods section, it is advisable to provide clear definitions for terms such as "Auto-injectables" and "Shortage of auto-injectables." Additionally, specify how the shortage of auto-injectables is defined according to Fimea. In the Discussion section, the following points are expected from the authors: a. Elaborate further on possible reasons for the findings related to the first and second research questions. b. Explore the association between 'Searching' and 'Shortage': specify the nature of this association, whether it might be temporal or a spurious correlation.

We look forward to receiving your revised manuscript.

Kind regards,

Muhammad Shahzad Aslam, Ph.D.,M.Phil., Pharm-D

Academic Editor

PLOS ONE

Journal Requirements:

Additional Editor Comments:

In the Methods section, it is advisable to provide clear definitions for terms such as "Auto-injectables" and "Shortage of auto-injectables." Additionally, specify how the shortage of auto-injectables is defined according to Fimea.

In the Discussion section, the following points are expected from the authors:

a. Elaborate further on possible reasons for the findings related to the first and second research questions.

b. Explore the association between 'Searching' and 'Shortage': specify the nature of this association, whether it might be temporal or a spurious correlation.

Reviewers' comments:

Reviewer's Responses to Questions

**Comments to the Author**

1. If the authors have adequately addressed your comments raised in a previous round of review and you feel that this manuscript is now acceptable for publication, you may indicate that here to bypass the “Comments to the Author” section, enter your conflict of interest statement in the “Confidential to Editor” section, and submit your "Accept" recommendation.

Reviewer #4: (No Response)

Reviewer #5: (No Response)

2. Is the manuscript technically sound, and do the data support the conclusions?

Reviewer #4: Yes

Reviewer #5: Partly

3. Has the statistical analysis been performed appropriately and rigorously? 

Reviewer #4: Yes

Reviewer #5: Yes

4. Have the authors made all data underlying the findings in their manuscript fully available?

Reviewer #4: Yes

Reviewer #5: Yes

5. Is the manuscript presented in an intelligible fashion and written in standard English?

Reviewer #4: Yes

Reviewer #5: Yes

6. Review Comments to the Author

Reviewer #4: -The contents of the research article are benefitial.

-There is no correct style of research writing. In general, it should attempt to be accurate, impersonal and objective. For example, personal pronouns like ‘We’ (informal language) are used less often than in other writing. So, some sentences need to be revised.

Reviewer #5: 1. At Line 72: Authors could write a more clear sentence. (? in 11.2% of shortage case......)

2. Line 78: It is not clear authors were intended to write "Auto-injectors" or "Epinephrine Auto-injectors".

3. Line 122-132: If it reflects the methods of search, should these lines put in 'Methods' Part?

4. Line 175, 176: If authors intend to justify inclusion of the study period in a sentence "The literature review found......", should it be part of Methods rather than of Purpose?

5. In Methods section: It is better to define a. Auto-injectables b. Shortage of auto-injectable. How Shortage of Auto-injectables is defined by Fimea?

6. The readers might not be friendly to interpret points from Table 2. Why many rows for a Auto-injectables like EpiPen is not clear.

7. In Discussion Part: It is expected the following points by authors:

a. Further elaboration of possible reasons for findings on research questions first and second.

b. Association between 'Searching' and 'Shortage': what type of association might be? Can it be temporal or spurious association?

7. PLOS authors have the option to publish the peer review history of their article (what does this mean?). If published, this will include your full peer review and any attached files.

Reviewer #4: No

Reviewer #5: **Yes: **Basant Adhikari

---

## [Author Response · Author response to Decision Letter 2]

24 Jan 2024

Dear Academic Editor Muhammed Shahzad Aslam, Basant Adhikari, and Reviewer #4, 

Thank you for your suggestions in order to improve our manuscript. We have now revised our paper by providing the answers below. We will be happy to do further revisions in the manuscript to move forward to publication in PlosOne. 

Kind regards,

Dr. Milla Mukka

Response to the Editor

Additional Editor Comments:

In the method section, it is advisable to provide clear definitions for terms such as "Auto-injectables" and "Shortage of auto-injectables." Additionally, specify how the shortage of auto-injectables is defined according to Fimea.

Response: Thank you for your comments. We have checked the correct terms in the literature, and we think that they are “auto-injectors” and “shortage of auto-injectors”. We have provided the definitions for them in the Materials and Methods section. Also, the definition from Finnish Medicines Agency has been provided in this section.

In the Discussion section, the following points are expected from the authors: 

a. Elaborate further on possible reasons for the findings related to the first and second research questions. 

b. Explore the association between 'Searching' and 'Shortage': specify the nature of this association, whether it might be temporal or a spurious correlation.

Response: We thank for these comments.

a. We have elaborated further on possible reasons for findings of research question one and two. First, more information on professionals’ searches has been provided in Discussion. Second, more information on the relation of citizens’ article openings and shortages, as well as summertime and lesser prescriptions, has been added in Discussion. 

b. We have now specified the association as “temporal relation” in Discussion and elsewhere in our paper. However, our data on searches and shortages are not able to provide statistical analysis, such as correlation, so we use “visual relation” or “temporal relation” between searches and shortages explored in the figures. We found temporal relation between searches and shortages, thus we mention “visually similar seasonal patterns” in Discussion. Also, this has been added in the Limitations section. 

Reviewer #4

The contents of the research article are beneficial. There is no correct style of research writing. In general, it should attempt to be accurate, impersonal and objective. For example, personal pronouns like ‘We’ (informal language) are used less often than in other writing. So, some sentences need to be revised.

Response: Thank you for this suggestion. We have now replaced active voice with passive voice regarding “we” and “our” (informal language) as suggested.

Reviewer #5

Note 1. At Line 72: Authors could write a more clear sentence. (? in 11.2% of shortage case......)

Response 1: Thank you for this comment. We have now revised the sentence as follows: “The wholesaler has reported the reason for the shortage to community pharmacies in 11.2% of medicine shortage cases.”

Note 2. Line 78: It is not clear authors were intended to write "Auto-injectors" or "Epinephrine Auto-injectors".

Response 2: Thank you for your comment. We have clarified the subtitle and text. 

Note 3. Line 122-132: If it reflects the methods of search, should these lines put in 'Methods' Part?

Response 3: Thank you for your comment. The literature search was initially done since it justifies our study because of the lack of prior knowledge and literature of epinephrine auto-injectors. The literature search was conducted according to the peer review comments in the first review round. Therefore, it was suggested that the literature search should be put in the section prior to the Methods part.

Note 4. Line 175, 176: If authors intend to justify inclusion of the study period in a sentence "The literature review found......", should it be part of Methods rather than of Purpose?

Response 4: Thank you for your comment. We have now clarified the Methods and Purpose part of the text by removing this sentence from Purpose to Methods. 

Note 5. In Methods section: It is better to define a. Auto-injectables b. Shortage of auto-injectable. How Shortage of Auto-injectables is defined by Fimea?

Response 5: Thank you for your comments. We have checked the correct terms in the literature, and we think that they are “auto-injectors” and “shortage of auto-injectors”. We have provided the definitions for them in the Materials and Methods section. Also, the definition from Finnish Medicines Agency was provided in this section.

Note 6. The readers might not be friendly to interpret points from Table 2. Why many rows for a Auto-injectables like EpiPen is not clear.

Response 6: Thank you for this comment. We have now clarified this by adding a footnote under the table. Shortage periods have now been merged in one cell in the table. Also, further information on this table has been added in the text. 

Note 7. In Discussion Part: It is expected the following points by authors:

a. Further elaboration of possible reasons for findings on research questions first and second.

b. Association between 'Searching' and 'Shortage': what type of association might be? Can it be temporal or spurious association?

Response 7: We thank for these comments.

a. We have elaborated further on possible reasons for findings of research question one and two. First, more information on professionals’ searches has been provided in Discussion. Second, more information on the relation of citizens’ article openings and shortages, as well as summertime and lesser prescriptions, has been added in Discussion. 

b. We have now specified the association as “temporal relation” in Discussion and elsewhere in our paper. However, our data on searches and shortages are not able to provide statistical analysis, such as correlation, so we use “visual relation” or “temporal relation” between searches and shortages explored in the figures. We found temporal relation between searches and shortages, thus we mention “visually similar seasonal patterns” in Discussion. Also, this has been added in the Limitations section.

---

## [Decision Letter · Decision Letter 3]

6 Feb 2024

Detecting epinephrine auto-injector shortages in Finland 2016‒2022: Log-data analysis of online information seeking

PONE-D-23-19109R3

Dear,

We’re pleased to inform you that your manuscript has been judged scientifically suitable for publication and will be formally accepted for publication once it meets all outstanding technical requirements.

Kind regards,

Muhammad Shahzad Aslam, Ph.D.,M.Phil., Pharm-D

Academic Editor

PLOS ONE

Additional Editor Comments (optional):

Reviewers' comments:

Reviewer's Responses to Questions

**Comments to the Author**

1. If the authors have adequately addressed your comments raised in a previous round of review and you feel that this manuscript is now acceptable for publication, you may indicate that here to bypass the “Comments to the Author” section, enter your conflict of interest statement in the “Confidential to Editor” section, and submit your "Accept" recommendation.

Reviewer #4: All comments have been addressed

Reviewer #5: All comments have been addressed

2. Is the manuscript technically sound, and do the data support the conclusions?

Reviewer #4: Yes

Reviewer #5: Yes

3. Has the statistical analysis been performed appropriately and rigorously? 

Reviewer #4: Yes

Reviewer #5: Yes

4. Have the authors made all data underlying the findings in their manuscript fully available?

Reviewer #4: Yes

Reviewer #5: Yes

5. Is the manuscript presented in an intelligible fashion and written in standard English?

Reviewer #4: Yes

Reviewer #5: Yes

6. Review Comments to the Author

Reviewer #4: Thank you for revising the acticle. Now, the article has been written in academic and grammarticle language with a good content.

Reviewer #5: (No Response)

7. PLOS authors have the option to publish the peer review history of their article (what does this mean?). If published, this will include your full peer review and any attached files.

Reviewer #4: **Yes: **Asst.Prof.Dr.Wissawa Aunyawong

Reviewer #5: **Yes: **Basant Adhikari

---

## [Editor Report · Acceptance letter]

29 Mar 2024

PONE-D-23-19109R3 

PLOS ONE

Dear Dr. Mukka, 

I'm pleased to inform you that your manuscript has been deemed suitable for publication in PLOS ONE. Congratulations! Your manuscript is now being handed over to our production team.

Kind regards, 

on behalf of

Dr. Muhammad Shahzad Aslam 

Academic Editor

PLOS ONE